# Network preservation reveals shared and unique biological processes associated with chronic alcohol abuse in NAc and PFC

Eric Vornholt[1,2]*, John Drake[3], Mohammed Mamdani[1], Gowon McMichael[1], Zachary N. Taylor[1], Silviu-Alin Bacanu[1,4], Michael F. Miles[1,5,6,7], Vladimir I. Vladimirov[1,3,8,9,10,11]*

**1** Virginia Institute for Psychiatric and Behavioral Genetics, Virginia Commonwealth University, Richmond, Virginia, United States of America, **2** Integrative Life Sciences Doctoral Program, Virginia Commonwealth University, Richmond, Virginia, United States of America, **3** Department of Psychiatry and Behavioral Sciences, Texas A&M University, Bryan, Texas, United States of America, **4** Department Psychiatry, Virginia Commonwealth University, Richmond, Virginia, United States of America, **5** VCU-Alcohol Research Center, Virginia Commonwealth University, Richmond, Virginia, United States of America, **6** Department of Pharmacology and Toxicology, Virginia Commonwealth University, Richmond, Virginia, United States of America, **7** Department of Neurology, Virginia Commonwealth University, Richmond, Virginia, United States of America, **8** Center for Biomarker Research and Precision Medicine, Virginia Commonwealth University, Richmond, Virginia, United States of America, **9** Department of Physiology & Biophysics, Virginia Commonwealth University, Richmond, Virginia, United States of America, **10** School of Pharmacy, Virginia Commonwealth University, Richmond, Virginia, United States of America, **11** Lieber Institute for Brain Development, Johns Hopkins University, Baltimore, Maryland, United States of America

* vornholtes@vcu.edu (EV); vivladimirov@tamu.edu (VIV)

**Data Availability Statement:** The mRNA and miRNA expression data are available on the NCBI GEO database. NAc data can be found at accession

## Abstract

Chronic alcohol abuse has been linked to the disruption of executive function and allostatic conditioning of reward response dysregulation in the mesocorticolimbic pathway (MCL). Here, we analyzed genome-wide mRNA and miRNA expression from matched cases with alcohol dependence (AD) and controls (n = 35) via gene network analysis to identify unique and shared biological processes dysregulated in the prefrontal cortex (PFC) and nucleus accumbens (NAc). We further investigated potential mRNA/miRNA interactions at the network and individual gene expression levels to identify the neurobiological mechanisms underlying AD in the brain. By using genotyped and imputed SNP data, we identified expression quantitative trait loci (eQTL) uncovering potential genetic regulatory elements for gene networks associated with AD. At a Bonferroni corrected p≤0.05, we identified significant mRNA (NAc = 6; PFC = 3) and miRNA (NAc = 3; PFC = 2) AD modules. The gene-set enrichment analyses revealed modules preserved between PFC and NAc to be enriched for immune response processes, whereas genes involved in cellular morphogenesis/localization and cilia-based cell projection were enriched in NAc modules only. At a Bonferroni corrected p≤0.05, we identified significant mRNA/miRNA network module correlations (NAc = 6; PFC = 4), which at an individual transcript level implicated miR-449a/b as potential regulators for cellular morphogenesis/localization in NAc. Finally, we identified eQTLs (NAc: mRNA = 37, miRNA = 9; PFC: mRNA = 17, miRNA = 16) which potentially mediate alcohol's effect in a brain region-specific manner. Our study highlights the neurotoxic effects of chronic alcohol abuse as well as brain region specific molecular changes that may impact the development of alcohol addiction.

no. GSE62699, and PFC data at accession no. GSE161999.

**Funding:** This work was supported by the National Institute on Alcohol Abuse and Alcoholism (https:// www.niaaa.nih.gov) and National Institute of Mental Health (https://www.nimh.nih.gov). Grant numbers: F31AA028180 (Eric Vornholt), R21AA022749/R01MH118239 (Dr. Vladimir I. Vladimirov), and P50AA022537 (Dr. Michael F. Miles). The funders had no role in study design, data collection and analysis, decision to publish, or preparation of the manuscript.

**Competing interests:** The authors have declared that no competing interests exist.

# Introduction

Alcohol use disorder (AUD) is a debilitating psychiatric illness with negative health, economic, and social consequences for nearly 15.1 million affected adults worldwide [1]. AUD risk is dependent upon both genetic and environmental factors, with a heritability of 0.49 [2]. The neurobiological framework for understanding how benign, recreational alcohol use leads to AUD follows various hypotheses [3–5], with the most commonly accepted being the cyclical model of addiction [6]. This hypothesis provides valuable insight into the functional specialization of different brain regions that underlie behavioral maladaptations associated with AUD [7]. However, the genetic architecture and molecular mechanisms contributing to alcohol-facilitated neuroadaptations remain widely unknown.

Postmortem brain studies provide the unique opportunity to interrogate neurobiological changes associated with addiction across brain regions and neural pathways [8, 9]. Among these, the mesocorticolimbic system (MCL), which connects the ventral tegmental area (VTA) to the prefrontal cortex (PFC), and nucleus accumbens (NAc), has proven especially sensitive to alcohol-associated neuroadaptations [10–12]. Recent postmortem brain studies of AUD have focused on examining gene and microRNA (miRNA) expression as the biological intermediate between genetic variation and molecular function [13–19]. Studying mRNA and miRNA interactions may also reveal functional relationships that mediate the differential expression of risk AUD genes based on the role miRNAs play in the destabilization and degradation of their target genes [20]. While single gene expression differences are continuously explored, network approaches, such as weighted gene co-expression network analysis (WGCNA), allows genes with correlated expression, and therefore likely related functions, to cluster into modules that then can be analyzed to identify dysregulated biological processes and molecular pathways associated with AUD [21]. Others and we have successfully implemented this method to identify gene networks associated with AUD within the MCL and other brain regions [16, 18]. While postmortem brain expression differences alone are insufficient to infer a causal relationship between AUD and neurobiological function, the integration of genetic information via expression quantitative trait loci (eQTL) analysis can help elucidate the regulatory mechanisms by which genetic variants associated with AUD impact gene expression [22].

Thus, in this study, we seek to expand upon previous research by jointly analyzing two key MCL areas, the NAc and PFC, to identify unique and shared neurobiological processes associated with alcohol dependence (AD). To achieve this, we utilize a case/control study design to identify genes and co-expressed gene networks associated with AD. We then performed a network preservation analysis to determine how well significant modules and their respective biological processes are conserved between the PFC and NAc of chronic alcohol abusers. Within the significant modules, we identified the most connected genes (termed hubs), which were then integrated with miRNA expression data analyzed using the same methodological framework. Finally, we assessed the genetic factors that might impact the functions of risk AD genes via expression quantitative trait loci (eQTL). The miRNA and eQTL analyses were performed in order to identify the regulatory mechanisms by which gene networks identified in PFC and NAc contribute to alcohol addiction.

# Materials and methods

## Tissue processing and RNA extraction

Postmortem brain tissue from 41 AD cases and 41 controls was provided by the Australian Brain Donor Programs of New South Wales Tissue Resource Centre (NSW TRC) under the

support of The University of Sydney, National Health and Medical Research Council of Australia, Schizophrenia Research Institute, National Institute of Alcohol Abuse and Alcoholism, and the New South Wales Department of Health [8]. Samples were excluded based on: (1) history of infectious disease, (2) circumstances surrounding death, (3) substantial brain damage, and (4) post-mortem interval > 48 hours. Total RNA was isolated from PFC (the superior frontal gyrus) and NAc tissue using the mirVANA-PARIS kit (Life Technologies, Carlsbad, CA) following the manufacturer's suggested protocol. RNA concentrations and integrity (RIN) were assessed via Quant-iT Broad Range RNA Assay kit (Life Technologies) and Agilent 2100 Bioanalyzer (Agilent Technologies, Inc., Santa Clara, CA) respectively. Samples were matched for RIN, age, sex (all male), ethnicity, brain pH, and PMI as part of a previous study [18] yielding a total of 18 case-control matched pairs (n = 36). Due to our matching, the RINs in PFC were slightly lower (mean = 4.5, ±2.04) compared to NAc (mean = 6.9, ±0.84). Previous reports, however, have demonstrated that in post-mortem brain studies reliable results are readily obtained even with RINs ≤4 [23]. For demographic information see S1 Table.

## Gene expression microarray and data normalization

Gene expression was assayed using Affymetrix GeneChip Human Genome U133A 2.0 (HG-U133A 2.0) on 22,214 probe sets spanning ~ 18,400 mRNA transcripts, and the Affymetrix GeneChip miRNA 3.0 microarray interrogating the expression of 1733 mature miRNAs as previously described [24]. None of the mRNA or miRNA probes were excluded based on quality control criteria outlined in previous studies [18]. Raw probe data were GCRMA background corrected, $\log_2$ transformed, and quantile normalized using Partek Genomics Suite v6.23 (PGS; Partek Inc., St. Louis, MO) to obtain relative gene expression values. A principal component analysis was used to identify potential outlier samples. Only one case sample was removed from the analyses, leaving 18 controls and 17 cases (n = 35) for both brain regions. It has become widely accepted to verify a subset of microarray-generated gene expression changes via an independent platform such as qPCR. Considering limited tissue availability and our extensive use of the Affymetrix platform in the past, we did not include microarray validation in this study which is similar to what other groups have done in the past [25]. We have previously 'validated' the same array and platform in independent qPCR experiments with a concordance between microarray and qPCR platforms exceeding 80% in the past [18].

## Analysis of differential gene expression

The relationship between AD case status and gene expression in PFC and NAc was analyzed via bidirectional stepwise regression for each gene. This approach is better suited to adjust for the confounding effect of covariates within each transcript's regression model than the robust linear regression approach employed previously in the analyses of NAc [18]. Regression coefficients were calculated in RStudio (ver. 1.66) using the Stats package (ver. 3.5.1). We further observed that brain pH, RIN, and neuropathology were the most influential covariates in the analyses of NAc expression data, while RIN and smoking history were the two most important covariates in the PFC expression analysis. Finally, we assessed proportion of variance explained by each covariate via the variancePartition package (ver. 1.20) [26]. For more details on our bidirectional stepwise regression, see S1 File.

## Network analyses

WGCNA was performed using the WGCNA package in RStudio (ver. 1.66). All nominally significant genes (p≤0.05) were used to generate a signed similarity matrix via pair-wise Pearson correlations. The nominal significance was chosen to (1) include genes with smaller effect

sizes, albeit true positive signals, (2) exclude genes with low disease variance, i.e., likely not associated with AD and (3) to provide a sufficient number of genes for the network analysis. WGCNA was performed as outlined previously by us and others [16, 18] and in S1 File. Module eigengenes (MEs), serving as a single aggregate expression value for each of the modules, were correlated to AD case-status and available demographic/biological covariates. To validate WGCNA module clustering, we performed a bootstrap based resampling of 100 iterations with replacement. Next, using WGCNA with the clusterRepro (ver. 0.9) package in RStudio, we identified the level of module preservation between the PFC and NAc by comparing adjacency matrices and calculating the composite preservation statistic ($Z_{summary}$). A $Z_{summary} > 10$ indicates strong evidence for network preservation, $Z_{summary} < 10 > 2$ indicates weak evidence of network preservation and $Z_{summary} < 2$ indicates no module preservation, as outlined previously [27].

## Gene set enrichment analysis

Gene set enrichment was performed using ShinyGo (ver. 0.61) gene annotation database [28]. Gene lists from the significant AD modules from NAc and PFC were enriched using GO biological processes consisting of 15,796 gene sets from the Ensembl BioMart release 96; all p-values for significantly enriched gene sets are FDR adjusted (FDR of 5%). We further performed cell type enrichment using the "*userListEnrichment*" option within the WGCNA package in R (ver. 1.66) as previously described [18]. Statistical significance of brain-list enrichment was determined via a hypergeometric test; all p-values were adjusted at FDR of 5%.

## Hub gene prioritization

Hub genes were defined based on the strength of intramodular connectedness, (also referred as module membership (MM)) calculated from the absolute value of the Pearson's correlation coefficient between module eigengene and expression values. Hub genes were prioritized for downstream analysis based on MM of $r \geq 0.80$ and a significant gene correlation with AD (at $p \leq 0.05$).

## eQTL analysis and GWAS/GTEx enrichment

DNA from the postmortem brain sample was processed and genotyped as part of a larger GWAS study [18]. Genotypes with excessive missingness (greater than 20%) and monomorphic for homozygous major and minor alleles were removed. We then selected only, local, cis-eQTLs, defined as SNPs 500kb from the start/stop positions for each hub gene. Such selected SNPs were pruned with Plink v1.9 to exclude variants in linkage disequilibrium ($R^2 \geq 0.7$). For eQTL detection, SNP effect on hub gene expression was analyzed via MatrixEQTL package (ver. 2.2) in R using a linear regression model adjusting for covariates. To identify potential disease risk eQTLs, we further tested for an interaction (SNP x AD) term between genotype and AD status using the "*modelLINEAR_CROSS*" argument. A significant genotype/disease interaction for a SNP/gene pair would indicate that the effect of genotype on expression is significantly different in AD cases versus controls. To determine the overlap between the eQTLs in our sample (at $p \leq 0.002$) and significant GWAS hits (at $p \leq 1E-4$) from previously reported alcohol and smoking GWAS [29, 30], we employed the Simes enrichment test [31]. We further tested the overlap between eQTLs obtained from our analyses against eQTLs obtained from GTEx consortium [32]. The significance of this overlap was assessed via a Fisher's exact test at $p \leq 0.05$ threshold. See S1 File for more details.

### MiRNA/mRNA target prediction

The relationship between significant miRNA and mRNA modules from each brain region was examined by performing a Pearson's correlation on the miRNA and mRNA module MEs using the Stats package (ver. 3.5.1) in RStudio. Significant miRNA/mRNA ME correlations (at FDR of 5%) were followed up with a more detailed series of analyses, in which individual mRNA hub and miRNA expression was correlated via Pearson's correlations using the miR-LAB package in R (ver. 1.14.3).

## Results

### AD case/control differentially expressed genes

When we assess each covariate's contribution to the overall gene expression variance, we see that the impact of a given covariate on gene expression is highly variable across individual genes. Thus, our approach to use a stepwise regression to select for the most influential set of covariates that contribute to the highest proportion of the variance and hence minimize model overfitting. The covariates showing the highest mean contribution to the variance were also the same factors that were most frequently incorporated into our regression models (S2 Table).

The bidirectional stepwise regression revealed 3,536 and 6,401 differentially expressed genes (DEG) in PFC and NAc, respectively, at the nominal p ≤0.05, of which 1,279 DEG were shared between the two regions. Among these, 603 and 494 genes were downregulated and upregulated, respectively, and 182 genes were expressed in opposite directions between the two regions. Within the DEG in NAc, nine genes (*ADH1B*, *ADH1C*, *H2AFZ*, *EIF4E*, *FTO*, *DRD2*, *SLC39A8*, and *VRK2*) were implicated in the largest and most recent AD GWAS [33]. At FDR of 5%, we identified 1,841 DEG from the NAc and 70 from the PFC. The miRNA regression analysis identified 430 and 170 nominally significant miRNAs in the NAc and PFC, respectively, with 168 miRNAs differentially expressed in NAc at FDR of 5% with no miRNA reaching FDR significance in PFC. To maintain an identical analytical pipeline for both brain regions and optimize the selection for the most influential confounding factors, we co-jointly analyzed the PFC expression data generated in this study with our previously published NAc expression data [18]. We observed a highly significant overlap between the differentially expressed genes identified in NAc from both studies (Fisher's exact test, p = 1E-10). For detailed information about regression coefficients, the regression models used for each transcript, partitioning of variance, and frequency of covariates incorporated into the analysis, see S2 Table.

### Gene network module clustering

In NAc, at a Bonferroni adjusted p ≤0.05, we identified 6 modules significantly correlated with AD case status (Fig 1A). Among these, NAc$_{darkgreen}$ was the only negatively correlated module, whereas NAc$_{darkorange}$, NAc$_{purple}$, NAc$_{magenta}$, NAc$_{skyblue}$, and NAc$_{greenyellow}$ were all positively correlated with AD cases relative to controls (Fig 1B). In PFC, we identified 3 modules significantly correlated to AD at Bonferroni adjusted p≤0.05 (Fig 1C). Of these, the PFC$_{pink}$ module was negatively correlated, while PFC$_{darkred}$ and PFC$_{lightgreen}$ were positively correlated with AD cases (Fig 1D). To assess the validity of these network modules, we performed a bootstrap resampling that showed consistent module clustering when compared to the original gene networks (S2 File).

### NAc and PFC network preservation

We performed a network preservation analysis to determine how well co-expressed networks from the PFC are conserved in NAc and vice versa. We focused primarily on the $Z_{summary}$ and

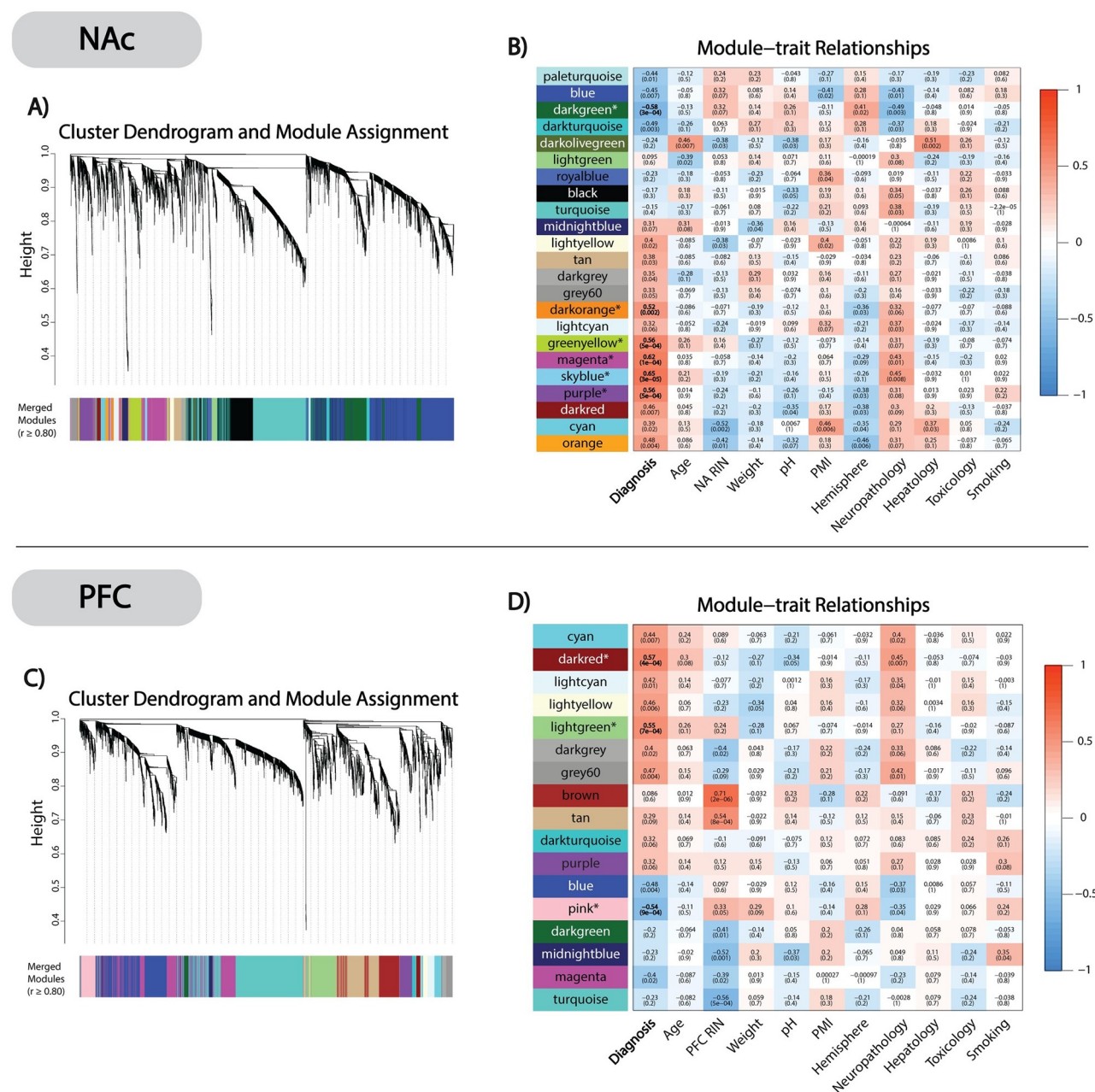

**Fig 1. WGCNA clustering and module-trait relationships. A)** NAc cluster dendrogram and module assignment with dissimilarity based on topological overlap. The 6,401 selected transcripts were clustered into 23 distinct modules. **B)** NAc module-trait relationship heatmap correlating (Pearson's) module MEs with AD diagnosis and covariates. Uncorrected p-values are given in parenthesis below each correlation coefficient. 6 AD associated significant modules (NAc$_{darkgreen}$, NAc$_{darkorange}$, NAc$_{greenyellow}$, NAc$_{magenta}$, NAc$_{skyblue}$, and NAc$_{purple}$) were identified after Bonferroni correcting p-values (* = p≤0.05). **C)** PFC cluster dendrogram and module assignment. The 3,536 selected transcripts were clustered into 17 different co-expressed modules. **D)** PFC module-trait relationship heatmap created as previously described. We identified 3 AD associated modules (PFC$_{pink}$, PFC$_{darkred}$, and PFC$_{lightgreen}$) after Bonferroni correcting p-values (* = p≤0.05).

*Median Rank* network preservation statistics because Z$_{summary}$ estimates network overlap by also taking into consideration network connectivity. *Median Rank* being invariant to module size, provides a more accurate estimate of network preservation since larger networks tend to be more conserved due to their size alone. We observed that NAc$_{darkorange}$ and NAc$_{purple}$

showed little to no network preservation ($Z_{summary}$ <2), NAc$_{skyblue}$, NAc$_{darkgreen}$, PFC$_{darkred}$, and PFC$_{pink}$ showed moderate levels of network preservation (2< $Z_{summary}$ <10), and NAc$_{greenyellow}$, NAc$_{magenta}$, and PFC$_{lightgreen}$ showed high levels of network preservation ($Z_{summary}$ >10) (Fig 2A and 2B). For detailed information about the individual density and connectivity statistics that were used to create the composite network preservation statistics, see the S3 Table.

## Identifying biological processes and cell-type enrichment of the AD significant modules

To gain perspective on the biological underpinnings of the significant gene networks from NAc and PFC, we performed a gene-set enrichment analysis, GO biological processes annotation (ShineyGO ver.61) and neuronal cell type enrichment for the two regions. As one of our

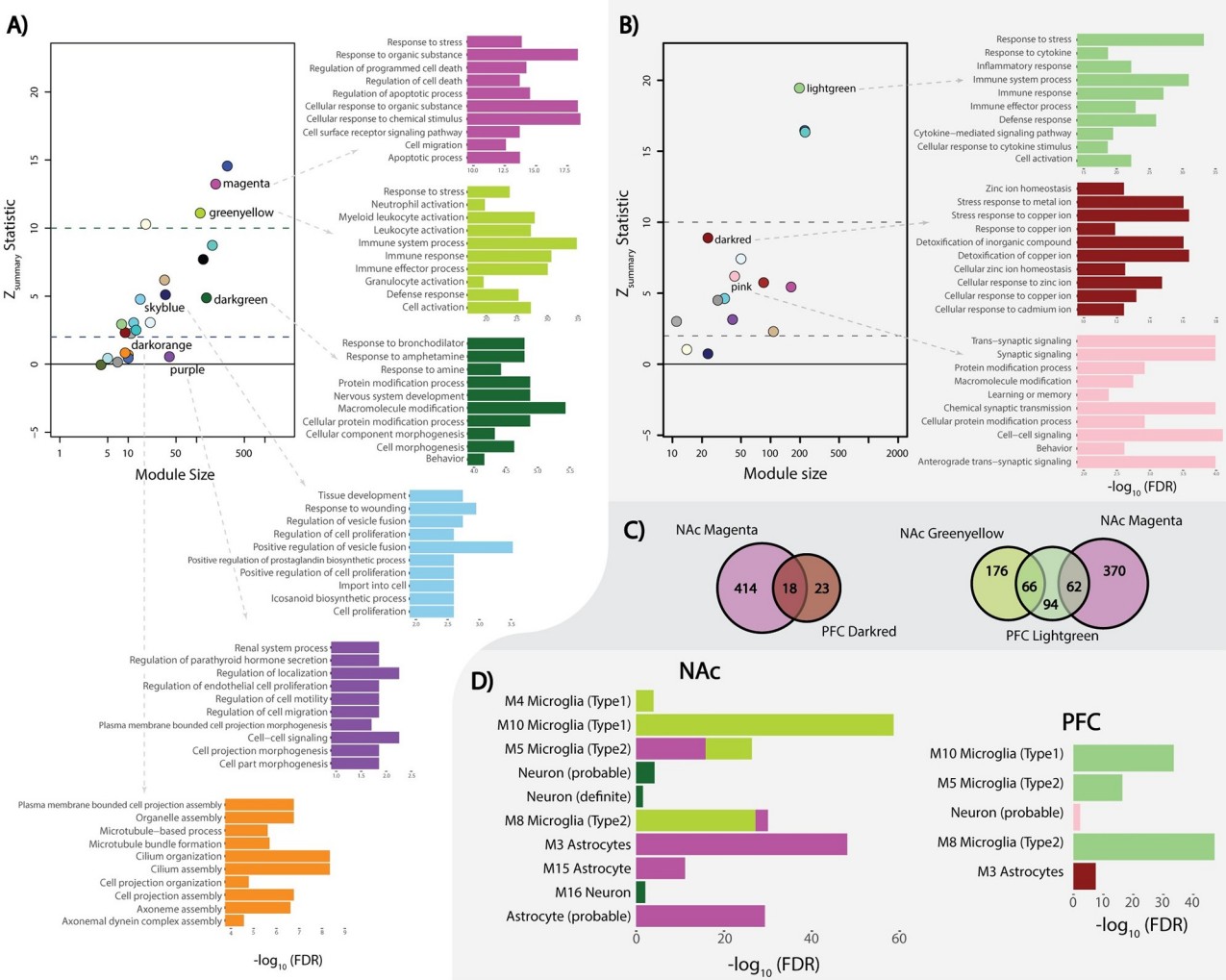

**Fig 2. Network preservation and gene-set enrichment.** *A)* NAc Z-summary statistic calculated as an aggregate of network preservation statistics (*Preservation level*: high = Z>10; moderate = 2<Z<10; low = Z<2) with color corresponded top-10 most significant (-log$_{10}$(FDR) transformed) GO biological processes for significant AD associated modules. *B)* PFC Z-summary statistic and corresponding GO biological processes term (-log$_{10}$(FDR) transformed). *C)* Venn-diagram of the shared transcripts from highly preserved NAc modules (NAc$_{magenta}$ and NAc$_{greenyellow}$) and their corresponding significant PFC modules (PFC$_{lightgreen}$ and PFC$_{darkred}$). *D)* Brain cell type gene-set enrichment from the NAc and PFC (-log$_{10}$(FDR) transformed). Colors correspond with their respective modules (NAc$_{greenyellow}$, NAc$_{magenta}$, NAc$_{darkgreen}$, PFC$_{pink}$, PFC$_{darkred}$, and PFC$_{lightgreen}$) with single gene sets enriched in modules.

aims was to identify unique and shared gene networks associated with AD in NAc and PFC, we focused our analyzes on NAc modules that were highly (i.e., NAc$_{greenyellow}$ and NAc$_{magenta}$) and poorly (i.e., NAc$_{darkorange}$, and NAc$_{purple}$) preserved in PFC. NAc$_{greenyellow}$ and NAc$_{magenta}$ are primarily associated with the immune response process (FDR ≤0.05) believed to be a consequence of neurotoxicity caused by chronic alcohol abuse (Fig 2A). These modules are enriched among microglia and astrocyte cell types (FDR ≤0.05), which is expected based on the functional properties of the glial cells (Fig 2D). The poorly preserved NAc modules showed enrichment within gene-sets associated with cilia-based cell projection and cell morphogenesis (FDR≤0.05) (Fig 2A).

Corollary, we performed gene-set enrichment analysis on PFC modules, which were highly and poorly preserved in NAc. (Fig 2B). Similar to the NAc$_{greenyellow}$ and NAc$_{magenta}$ modules, the highly preserved PFC$_{lightgreen}$ module was associated with immune response processes (FDR ≤0.05) and significant microglial cell type enrichment (FDR ≤0.05) (Fig 2D). PFC$_{darkred}$ and NAc$_{magenta}$, were moderately preserved with each other (Fig 2C) with PFC$_{darkred}$ showing astrocyte cell type enrichment (Fig 2D). Interestingly, a class of genes in one family of immune response proteins, metallothioneins (MTs), contained in both the PFC$_{darkred}$ and NAc$_{magenta}$ modules, were differentially expressed in both brain regions between cases and controls (Fig 3).

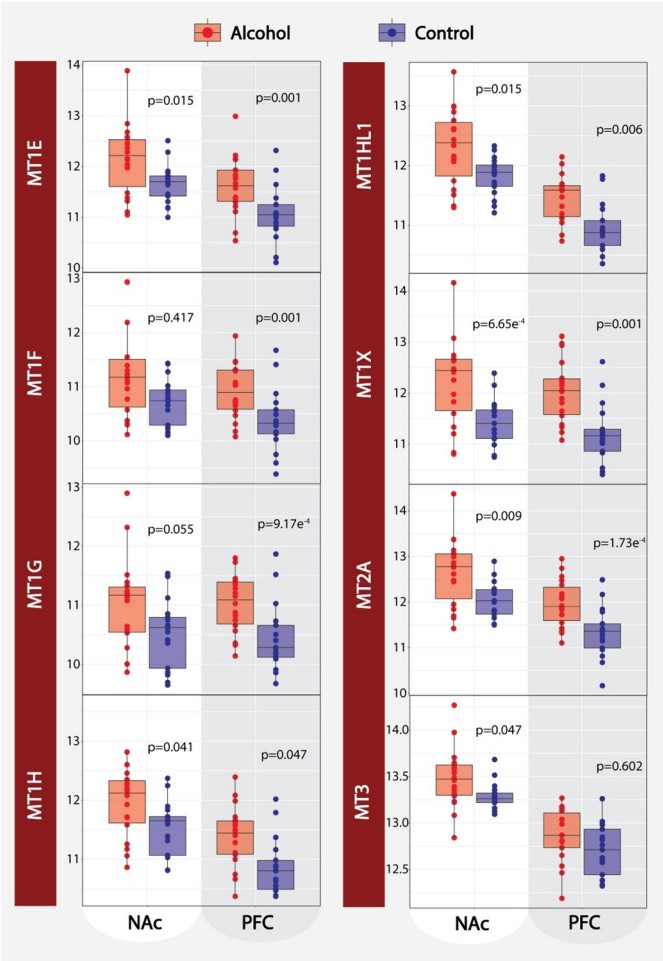

**Fig 3. Metallothionein gene expression.** Relative expression of 8 metallothionein cluster genes (*MT1E*, *MT1F*, *MT1G*, *MT1H*, *MT1HL1*, *MT1X*, *MT2A*, and *MT3*) comparing AD case to controls for both the NAc and PFC. P-values presented for each transcript are based on our bidirectional stepwise regression.

For the complete gene set enrichment analyses for all modules associated with AD in NAc and PFC see S4 Table. Since hubs are considered the most important genes for preserving the network's integrity, when these analyses were further limited only to the hub genes, not surprisingly, we captured the same GO terms and biological process that we observed from the entire module gene lists (S4 Table).

## Hub genes of potential biological significance

To identify candidate hub genes of potential biological significance, we focused on the relationship between intramodular connectivity (i.e., module membership (MM)) and gene significance (GS) to AD case status (S1 File). Of the 459 genes from the 3 significant PFC modules and the 6 significant modules in NAc, we identified 99 and 433 unique hub genes with MM ≥0.80, respectively. We focus on the hub genes due to their biological relevance to AD and predicted role as drivers of expression for the entire module [34]. For full PFC and NAc transcript information regarding individual MM and GS see S5 Table.

## Detection of miRNA gene network modules in NAc and PFC

In NAc and PFC, we identified miRNA modules with varying levels of significant correlation to AD case status. The NAc miRNA data revealed 430 nominally significant loci, which clustered in 5 modules ranging from 18 (NAcmi$_{green}$) to 259 (NAcmi$_{turquoise}$) loci in size, of which, at Bonferroni adjusted p ≤0.05, three miRNA modules remained significantly correlated to AD (NAcmi$_{yellow}$, NAcmi$_{brown}$, and NAcmi$_{turquoise}$). Of these, NAcmi$_{yellow}$ and NAcmi$_{brown}$ were negatively correlated, whereas NAcmi$_{turquoise}$ was positively correlated within AD (Fig 4A). The 170 miRNA transcripts from the PFC clustered into 6 modules ranging in size from 9 (PFCmi$_{red}$) to 55 miRNA transcripts (PFCmi$_{turquoise}$), of which PFCmi$_{yellow}$ and PFCmi$_{red}$, remain significant at Bonferroni adjusted p ≤0.05; both miRNA modules were negatively correlated with AD (Fig 4D). For further details on the miRNA WGCNA from both brain regions, including individual transcript MM and GS values, please refer to S6 Table.

## MiRNA networks show unique patterns of regulation

In an attempt to identify a higher order system, network levels of interactions, existing between the AD significant mRNA and miRNA modules we correlated their respective module MEs. From the NAc, we identified 2 significant positive mRNA/miRNA ME correlations and 4 negative ME correlations at Bonferroni adjusted p ≤0.05 (Fig 4B). To better understand the biological function of miRNA/mRNA interacting networks at specific loci, we honed on the interaction between individual miRNA/gene pairs. After correlating individual mRNA hubs and miRNA, we identified 1,801 significant mRNA/miRNA interactions (FDR ≤0.10) spanning 318 genes and 68 miRNA loci (S7 Table). Interestingly, we observed 97% (35/36) of the purple mRNA module hub genes to be negatively correlated with either mir-449a or mir-449b from NAcmi$_{brown}$ (Fig 4C). In PFC, we identified one positive mRNA/miRNA ME correlation and 3 negative correlations at Bonferroni adjusted p≤0.05 (Fig 4E). Individual mRNA/miRNA interaction analysis from the PFC revealed 6 mRNA/miRNA interactions (FDR of ≤0.10) spanning 6 genes and one miRNA transcript, mir-485-5p. For a full list of individual mRNA/ miRNA interactions, see S7 Table.

## Brain region specific eQTL regulation of differential gene expression

In NAc, we detected a total of 36 mRNA eQTLs spanning 17 unique genes and 9 miRNA eQTLs covering 4 different miRNA (FDR ≤0.10). Of the 17 hubs with significant eQTLs, 7 are from

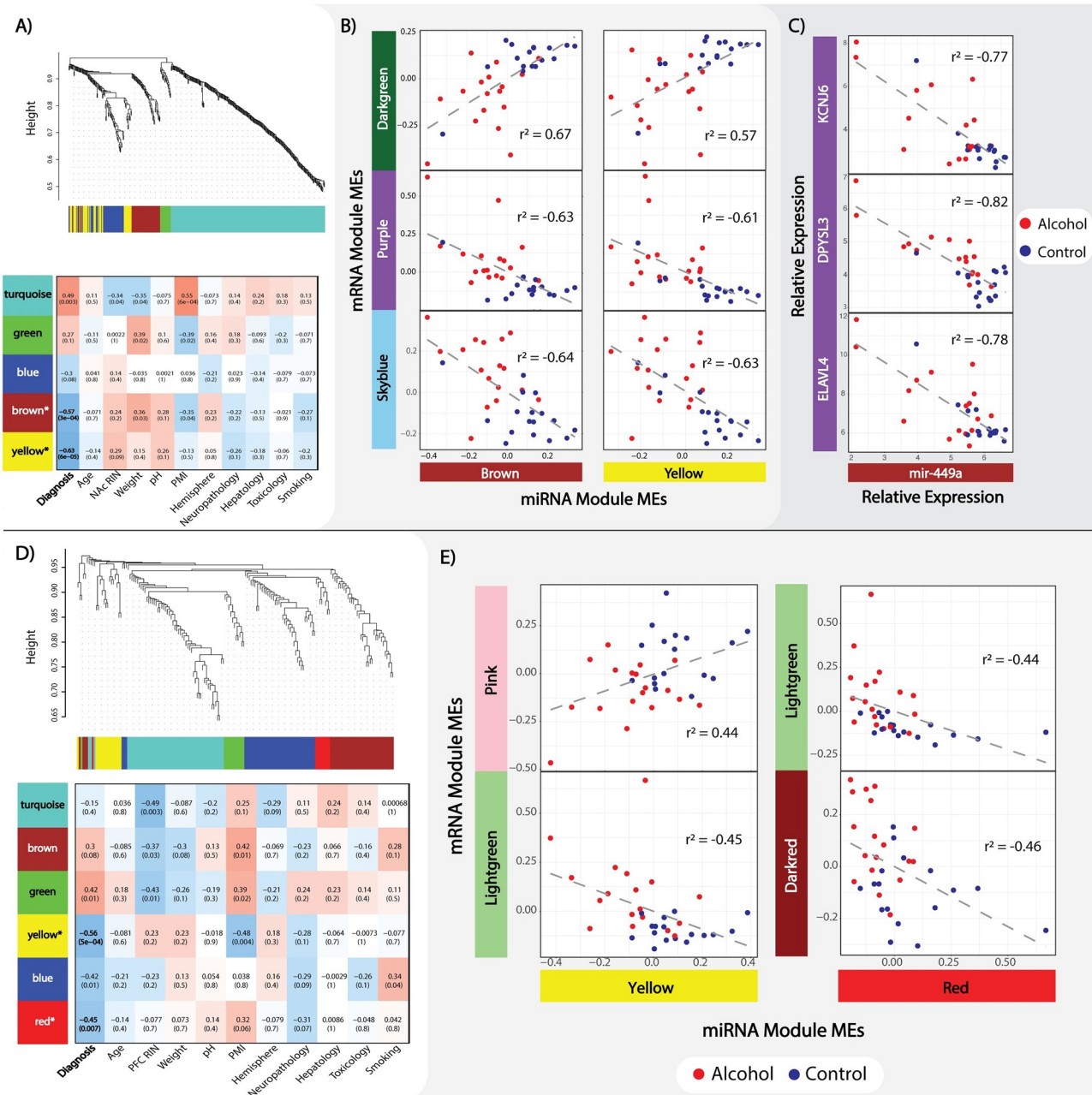

**Fig 4. MiRNA WGCNA and mRNA:miRNA interaction.** *A)* NAc miRNA cluster dendrogram and module assignment with module-trait relationship heatmap, both as previously described in Fig 1. *B)* Bonferroni adjusted significant (p≤0.05) NAc mRNA/miRNA module ME correlations (Pearson's). Alcohol and control groups are separated by color to emphasize sample clustering. *C)* Significant (FDR≤0.05) correlation (Pearson's) between mir-449a and selected mRNA transcripts from the low network preserved NAc$_{purple}$ module. *D)* PFC miRNA cluster dendrogram module assignment along with module-trait relationship heatmap. *E)* Bonferroni adjusted significant (p≤0.05) NAc mRNA:miRNA module ME correlations (Pearson's).

NAc$_{darkgreen}$ (*VRK1, INPP4A, HMP19, DKK3, PCDH8, RNF34*, and *RASGRP1*), 4 from NAc$_{greenyellow}$ (*FCGR3A, CTSS, AASS*, and *RNASE4*), 3 from NAc$_{darkorange}$ (*DNALI1, CCDC81*, and *SPAG6*), 2 from NAc$_{purple}$ (*HIVEP1* and *GNAS*), and one from NAc$_{magenta}$ (*VAMP5*). Within the PFC we identified 34 eQTLs spanning 16 unique genes and 18 miRNA covering 7 different miRNA transcripts (FDR ≤0.10). Of these, 11 genes are from PFC$_{lightgreen}$ (*SERPINH1*,

*CDKN1A*, *PNP*, *EMP1*, *FKBP5*, *IL4R*, *TNFRSF10B*, *RTEL1/TNFRSF6B*, *SERPINA1*, *MAFF*, and *SERPINA2*) and 5 from PFC$_{pink}$ (*GAD2*, *ACTL6B*, *KCNF1*, *SEZ6L*, and *EFNB3*). Among our significant eQTLs, we highlight two examples: *FCGR3A*:rs12087446 (NAc p = 3.24E-07; PFC p = 0.002) from the highly conserved NAc$_{greenyellow}$ module and *DNALI1*:rs12119598 (NAc p = 1.94E-09; PFC p = 0.150) from the poorly conserved NAc$_{darkorange}$ module. The brain region specific eQTL impact on the expression of these two genes suggests that different genetic mechanisms are likely at play in NAc and PFC that may further shed light on the different behavioral measures encoded by the two brain regions (Fig 5). For the full list of cis-eQTL, please refer to S8 Table. To highlight the potential clinical importance of our findings and provide functional support for previous genetic studies, we also tested for enrichment of our clinically relevant eQTLs (i.e., testing only SNPs that showed a significant (SNP x AD) interaction term) and previously published GWAS of addiction phenotypes. While the overlap did not reach formal significance, likely due to the smaller GWAS sample size, we nevertheless observed suggestive enrichment, i.e., GSCAN drinks per week p = 0.195; GSCAN smoking initiation p = 0.251;

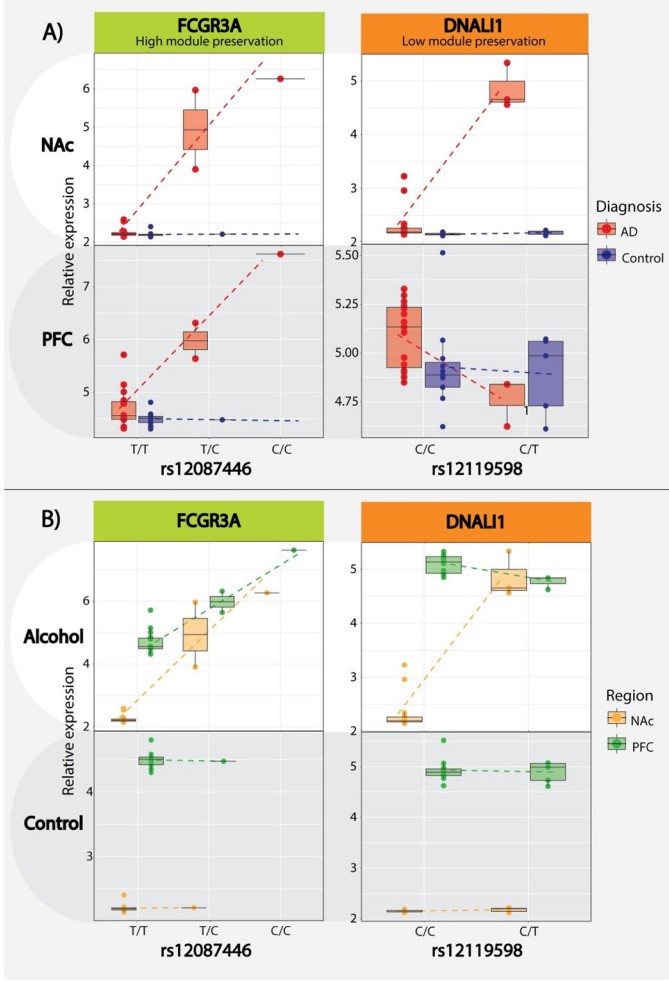

**Fig 5. Cis-eQTL analysis. A)** Cis-eQTL boxplot directly comparing AD case/control designation with the *FCGR3A: rs12087446* eQTL from the high network preservation NAc$_{greenyellow}$/PFC$_{lightgreen}$ module and the *DNALI1:rs12119598* eQTL from the low preservation NAC$_{darkorange}$ module, the relative expression is presented on the y-axis and SNP/ genotype on the x-axis. **B)** Alternative boxplot visualization of the same cis-eQTL directly comparing differences between brain regions.

GSCAN smoking cessation p = 0.147; and COGA+Irish p = 0.299. Finally, we attempted to replicate all eQTLs in our study, irrespective of their potential disease relevance, in the GTEx database using the Fisher's exact test. Interestingly, we observed a significant overlap between our eQTLs detected in the PFC (n = 2,368, 6.6% of eQTLs tested, p = 0.003), but not in the NAc (n = 5,436, 3.4% of eQTL tested, p = 1).

## Discussion

AUD continues to be a growing public health concern with a complex and poorly understood etiology as recreational alcohol use becomes habitual and problematic. The broad goal of this study is to identify the neurobiological processes associated with chronic alcohol use via analyzing brain region-specific gene networks from the NAc and PFC. To understand the human behavior leading to addiction, it is important to investigate how chronic alcohol use impacts expression changes in the evolutionarily newer cortical areas, in contrast to the older, more evolutionarily conserved subcortical brain regions [35]. Here, we attempt to understand the neurobiological underpinnings of alcohol specific reward conditioning in the NAc and disruption of executive function within PFC [6] through identifying gene networks and biological processes associated with AD that are conserved or unique to each brain region. Additionally, we assessed the relationship between the miRNA and mRNA networks significantly correlated to AD based on the miRNA functions to induce mRNA degradation and/or translational inhibition. Finally, we tested the impact of genetic variants on gene expression in a disease dependent manner via AD-mediated eQTL analysis.

Our network analyses are consistent with previously published reports by others and us, showing the upregulation of immune response mechanisms among AD cases as a byproduct of alcohol's neurotoxic effects [36]. The immune-related modules show significant enrichment for both astrocyte and microglial cell types, which has been validated by previous alcohol studies and the known immune functions of astrocyte and microglia in the brain [37, 38]. More importantly, we observed generalized up-regulation of immune response mechanisms within both the PFC and NAc, suggesting that the neurotoxic response to chronic alcohol use is ubiquitous across cortical and subcortical brain regions. Interestingly, in both brain regions, we further identified differentially expressed genes in the metallothionein cluster (*MT1HL1*, *MT1H*, *MT1X*, *MT1E*, *MT1G*, *MT1F*, *MT2A*, and *MT3*). The metallothionein cluster is primarily responsible for maintaining the cellular homeostasis of zinc and copper while also regulating oxidative stress [39]. Zinc is an essential catalytic cofactor for alcohol metabolism via alcohol dehydrogenase [40]. Free or "chelated" zinc ions ($Zn^{2+}$) are seen in abundance in the brain, specifically at ionotropic glutamate receptors such as the NMDA receptor family. The interaction between $Zn^{2+}$ and NMDAR activity has shown to be an important contributor to synaptic plasticity through regulating postsynaptic density assembly [41]. It is well understood that chronic alcohol abuse leads to varying degrees of organ-wide zinc deficiency [42]; however, the neurobiological consequences of how zinc deficiency in the brain contributes to AD neuropathology is poorly understood. We believe this interaction between chronic alcohol abuse, metallothionein expression, zinc deficiency, and synaptic plasticity is an important avenue for future research that should be explored.

In addition to identifying dysregulated immune response mechanisms, we validate recent studies showing differential expression among signaling and neurodevelopmental processes within AD cases [13, 15, 16, 18]. However, these processes are less conserved between cortical and subcortical regions, likely due to the different neuronal composition and functional properties of the PFC and NAc [43]. Interestingly, two NAc modules that primarily associate with cilium assembly (NAc*darkorange*) and cellular localization/morphogenesis (NAc*purple*) show

limited network preservation within the PFC. There has been increasing evidence suggesting primary cilia aid in facilitating extrasynaptic signaling during adult neurogenesis [44, 45], an important aspect of addiction related extracellular membrane plasticity [46]. For example, *GRP88*, a g-protein coupled receptor and primary cilia enriched gene [47], was linked to increased alcohol seeking behaviors in knock out (KO) mice models [48], further reinforcing the importance of primary cilia in AUD etiopathology. The cilium assembly genes enriched in NAc$_{darkorange}$, were shown to be associated with axonemal dynein assembly (*DNAAF1*, *DNAI2*, and *DNALI1*). A recent gene expression study in adolescent rat hippocampus identified increased expression of two dynein associated genes (*dnai1* and *dnah5*) [49]. One explanation for increased expression of primary cilia associated genes in the NAc relative to the PFC is related to potential discrepancies in adult neurogenesis between subcortical vs cortical brain regions. It is well understood that most adult neuronal stem cells originate in the ventricular–subventricular zones (V-SVZ) and migrate to adjacent cortical and subcortical brain regions as neuroblasts to promote neurogenesis [50]. A recent study showed increased adult neurogenesis of medium spiny neurons within the NAc and that the migration and incorporation of new neurons was experience-based [51]. We believe that the increased expression of genes that encode for the cilia assembly complex may reflective of experience mediated neurogenesis of medium spiny neurons in NAc, except being driven by chronic alcohol consumption instead of pain. These new neurons formed in response to alcohol use may play an important role in the reward response deficits we often associate with addiction and AUD [4].

Other interesting findings arise from our mRNA/miRNA interactions, e.g., when correlating the MEs from mRNA and miRNA modules, we see distinct patterns between cases and controls within both brain regions. Based on the known function of miRNAs in regulating the expression of target mRNAs [52] we can infer these significant miRNA networks may serve as a driving contributor for differential network expression between AD cases and controls. Specifically, 97% (35/36) of the hub genes from the NAc$_{purple}$ module were significantly negatively correlated with either mir-449a or mir-449b. Mir-449a/b have primarily been studied in the context of spermatogenesis and cellular proliferation in cancer [53–55]. Based on the mRNA-miRNA correlations, our study suggests that mir-449a/b cluster has additional functions related to cellular proliferation in the brain. Among the genes correlated with mir-449a in the NAc, *ELAVL4*, *DPYSL3*, and *KCNJ6* have shown significant associations with AD in other expression, and genetic association studies [19, 56, 57], as well as being implicated in other substance use disorders [58–61].

In an attempt to understand the causal nature of the gene networks associated with AD, we integrated genetic information via eQTL analysis. We were able to detect a significant number of mRNA and miRNA cis-eQTLs from both brain regions. We selected highly significant eQTLs (*FCGR3A* (Fc fragment of IgG receptor IIIa):*rs12087446* and *DNALI1* (dynein axonemal light intermediate chain 1):*rs12119598*) based on *FCGR3A* and *DNALI1*'s role as network hubs to highlight the interaction between AD case status and eQTL while also demonstrating brain region-specific eQTL variation. *FCGR3A* is one of the low-affinity Fc receptor genes important for NK cell-mediated antibody-dependent cytotoxicity [62] and a hub gene from our highly conserved NAc$_{greenyellow}$ and PFC$_{lightgreen}$ modules. The consistent effect of *rs12087446* on *FCGR3A* expression between both brain regions suggests the genetic impact on immune response processes might also be ubiquitous across the brain of chronic alcohol users. Differential FCGR3A expression was recently shown to be associated with both alcohol preference and binge-like behaviors in the ventral tegmental area of rats [63]. In contrast, *DNALI1*, a hub gene in the cilium assembly enriched NAc$_{darkorange}$ module, is under the genetic control of specific eQTL only in NAc but not in PFC, suggesting that changes to cilia organization due to alcohol abuse might be under different genetic control between the two brain regions. We observed suggestive

evidence for enrichment between our eQTLs and previously published GWAS of alcohol or other addiction phenotypes, such as smoking. We believe this is primarily due to three factors: 1) low statistical power within our sample to detect genetic signals that would otherwise appear in large-scale GWAS studies, 2) our selective study design focusing only on potentially clinically relevant eQTLs, and 3) the presence of variants with a lower MAF in the GWAS potentially not detectable in our dataset. We further successfully replicated our eQTLs in the GTEx database for PFC, but not NAc. One possible explanation is that the increased number of DEG in the NAc relative to PFC with the fact GTEx does not include AD diagnosed brains in their analyses [64] effectively limits our ability to replicate GTEx eQTLs based on significant and potential subtle non-significant expression changes among AD cases.

## Conclusion

The strength of this study lies in our ability to compare and contrast expression changes between subjects with AD and controls within two different brain regions. We successfully identified gene networks and biological processes from both brain regions that were validated by previous AD studies as well implicated a novel biological process (cilia assembly) and gene family (metallothionein cluster) as potentially important for the development of AD. Our mRNA/miRNA interaction analysis pinpointed mir-449a/b cluster as an important regulator of differentially expressed genes between AD cases and controls. Finally, via our eQTL analysis, we provided evidence that mRNA and miRNA expression differences between AD cases and controls might be under brain region specific genetic control. While our sample size could be perceived as a limitation, we mitigated this by utilizing WGCNA to aggregate differentially expressed genes into biologically relevant modules with single expression values, effectively increasing our power to detect significant AD associations within a multivariate framework. Additionally, to increase the power of our study, considering the more prevalent and heavier drinking patterns in men, we assessed the molecular processes of alcohol drinking in male subjects only. While we recognize the importance of comparing the molecular pathology of drinking between the two sexes, we would like to highlight observations from genetic epidemiological studies showing male and female subjects to have a similar genetic predisposition to alcohol abuse [65]. We further recognize that a number of our significant AD associated modules in PFC were also nominally correlated to neuropathology (p≤0.05). This is not entirely unexpected, given the known neuropathological impact of chronic alcohol abuse [66]. Finally, while we understand that the lower RINs from the PFC can be seen as confounding factor, studies have suggested that reliable data can still be obtained from postmortem brain tissue even with suboptimal RNA quality [67–69]. However, our careful analytical design to adjust for the impact of RIN on gene expression maintains the robustness of our results even in the presence of lower RINs.

Overall, the broader impact of our findings is the understanding that chronic alcohol consumption can reinforce addiction behaviors through dysregulating different biological process across various brain regions. This information could potentially lead to more focused therapies for AUD by targeting important brain regions specific neurobiological pathways involved in the development of alcohol addiction. While our results point to certain biological processes that differentiate between the PFC and NAc, these findings require replication in an independent postmortem brain samples spanning other cortical and subcortical brain regions. Additional support for the postmortem brain findings presented here can also be obtained by studying ethanol activity in animal models or neuronal cell cultures. Increased research within the methodological framework of our study can help validate our findings and identify biological processes and genes that play the most significant role in the development of AUD.

## Supporting information

**S1 File. Expanded methodology descriptions (bi-directional stepwise regression, WGCNA, addiction GWAS enrichment, and GTEx eQTL replication) and supplemental figure displaying MM by gene significance for AD associated mRNA and miRNA modules.**
(DOCX)

**S2 File. 100 permutation robust WGCNA dendrogram clustering.**
(PDF)

**S1 Table. Sample demographics.**
(XLSX)

**S2 Table. Stepwise regression coefficients, models, covariate frequency counts and variance partitioning (VP).** 3.1) mRNA, 3.2) miRNA.
(ZIP)

**S3 Table. Network preservation Z-summary table and supplemental network preservation statistics.**
(XLSX)

**S4 Table. Full GO biological processes annotation for each AD associated module from the NAc and PFC.** 6.1) NAc, 6.2) PFC.
(ZIP)

**S5 Table. mRNA WGCNA module membership (MM) and gene significance(GS) values with AD associated modules isolated in separate tabs.** 7.1) NAc, 7.2) PFC.
(ZIP)

**S6 Table. miRNA WGCNA module membership (MM) and gene significance(GS) values with AD associated modules isolated in separate tabs.** 8.1) NAc,8.2) PFC.
(ZIP)

**S7 Table. Top mRNA/miRNA correlations (NAc = top 2000; PFC = Top 500).**
(XLSX)

**S8 Table. cis-eQTL analysis results.**
(XLSX)

## Acknowledgments

We would like to acknowledge the participants who made this study possible as well as the Australian Brain Donor Programs of New South Wales Tissue Resource Centre (NSW TRC) as part of The University of Sydney, National Health and Medical Research Council of Australia, Schizophrenia Research Institute, National Institute of Alcohol Abuse and Alcoholism, and the New South Wales Department of Health for providing us with this invaluable resource.

## Author Contributions

**Conceptualization:** Vladimir I. Vladimirov.

**Data curation:** John Drake.

**Formal analysis:** Eric Vornholt, Silviu-Alin Bacanu.

**Funding acquisition:** Eric Vornholt, Michael F. Miles, Vladimir I. Vladimirov.

**Investigation:** Eric Vornholt, Mohammed Mamdani, Gowon McMichael.

**Project administration:** Vladimir I. Vladimirov.

**Supervision:** Vladimir I. Vladimirov.

**Visualization:** Eric Vornholt.

**Writing – original draft:** Eric Vornholt, Michael F. Miles.

**Writing – review & editing:** John Drake, Zachary N. Taylor, Vladimir I. Vladimirov.

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
