## [Decision Letter · Decision Letter 0]

28 Oct 2020

PONE-D-20-30169

Network Preservation Reveals Shared and Unique Biological Processes Associated with Chronic Alcohol Abuse in NAc and PFC.

PLOS ONE

Dear Dr. Vornholt,

Thank you for submitting your manuscript to PLOS ONE. After careful consideration, we feel that it has merit but does not fully meet PLOS ONE’s publication criteria as it currently stands.

You will see the detailed comments of the two Reviewers below. However, I am particularly concerned with the potential RNA degradation in the PFC samples, as mentioned by Reviewer 2 . Differential degradation could impact brain regional analysis. I hope you can adequately address this concern. If you can, I also suggest to add a more generalized take-home message to Discussion section.

If you feel that are able to adequately address these main concerns, I invite you to submit a revised version of the manuscript that addresses all points raised during the review process.

If you decide to do so, please submit your revised manuscript by Dec 11 2020 11:59PM. If you will need more time than this to complete your revisions, please reply to this message or contact the journal office at plosone@plos.org. Please include the following items when submitting your revised manuscript:

We look forward to receiving your revised manuscript.

Kind regards,

Andrey E Ryabinin, Ph.D.

Academic Editor

PLOS ONE

Journal Requirements:

Reviewers' comments:

Reviewer's Responses to Questions

**Comments to the Author**

1. Is the manuscript technically sound, and do the data support the conclusions?

Reviewer #1: Yes

Reviewer #2: Partly

2. Has the statistical analysis been performed appropriately and rigorously? 

Reviewer #1: Yes

Reviewer #2: Yes

3. Have the authors made all data underlying the findings in their manuscript fully available?

Reviewer #1: Yes

Reviewer #2: Yes

4. Is the manuscript presented in an intelligible fashion and written in standard English?

Reviewer #1: Yes

Reviewer #2: Yes

5. Review Comments to the Author

Reviewer #1: Rigorous and interesting analysis of mRNA and miRNA expression associated with alcohol dependence in post-mortem male human brain.

Methods.

Very detailed and well described methods.

Might be interesting to the reader to comment on which covariates had a larger effect on gene expression.

S1 Table. Probably need better legend. What do the numbers mean for Hemisphere, Neuropathology, Hepathology (#should be hepatology?), Toxicology, Smoking.

Results.

Figure 3. Significance levels for DE not noted in figure, legend, or text.

Hub genes of potential biological significance. This section is less interesting. Are they enriched for relevant biological processes or GWAS hits? Are they sig. DE between cases and controls? Other than their high degree of connectivity within modules enriched for AD, what is there biological significance? Or maybe integrate into another results section?

Figure 5. Units of expression not clearly described in the legend.

Discussion.

Further discussion of the interesting finding of cilium assembly in the NAc but not in the PFC may be warranted. Adult neurogenesis in the NAc? Neurogenesis of medium spiny neurons in the nucleus accumbens continues into adulthood and is enhanced by pathological pain. García-González et al., Mol. Psych. 2020. PMID: 32612250

Reviewer #2: The report by Vornholt et al. is a descriptive study that utilizing genome-wide mRNA and miRNA expression from postmortem brain of AD cases compared to matched controls. Gene co-expression networks, mRNA/miRNA interactions, and eQTL analyses were used to identify expression patterns that are uniquely and differently dysregulated in the PFC and NAc. The analytical pipelines are well described and mostly robust.

The Methods indicates 42 cases and 42 controls. Then after matching, 72 total cases. Table S1 shows demos for 70 cases (1 case was a statistical outlier). It is not clear why the initial 12 cases were excluded from the original group (84).

It is not clear from the beginning of the Methods section that the NAc data were reported previously by the same group. Utilization of the previously published data was clarified in the first section of the Results but needs to be outlined clearly in the Methods. Also, for clarity, any annotation updates and data handling differences for the dataset should be listed since the data were published in 2015.

There is significant concern about the within subject RNA quality differences for each of the brain regions. A major goal of the reported work was to identify overlapping and unique expression differences in PFC and NAc. There are clear differences in RNA quality between the 2 brain regions. This is likely degradation due to the sample processing at different times. The authors argue reliable data can be obtained from RINs<=4; however, the impact of RNA degradation on downstream expression analysis can be significant, especially when trying to compare multiple brain regions from the same individual. The cross-region analysis is likely compromised.

The major components of the study (descriptive statistics, WGCNA analysis, network preservation, mRNA/miRNA interactions) are potentially interesting; however, there is not a well-integrated overview or thematic model presented from consolidated findings. It is especially hard to understand exactly what the reader will gather from complex mRNA/miRNA interactions.

Functional validation of key findings would significantly strengthen the study. At a minimum, testable hypotheses should be discussed based on the integrated analyses.

6. PLOS authors have the option to publish the peer review history of their article (what does this mean?). If published, this will include your full peer review and any attached files.

Reviewer #1: No

Reviewer #2: No

---

## [Author Response · Author response to Decision Letter 0]

2 Nov 2020

Editors Comment: You will see the detailed comments of the two Reviewers below. However, I am particularly concerned with the potential RNA degradation in the PFC samples, as mentioned by Reviewer 2 . Differential degradation could impact brain regional analysis. I hope you can adequately address this concern. If you can, I also suggest to add a more generalized take-home message to Discussion section.

We appreciate Editor’s feedback and hope to have addressed the comment in respect to PFC RINs below in our response to Reviewer #2. Additionally, we have added a more generalized take-home message at the end of the discussion (pg. 21; line 534-538)

Reviewer #1: Rigorous and interesting analysis of mRNA and miRNA expression associated with alcohol dependence in post-mortem male human brain.

Methods.

Very detailed and well described methods.

Might be interesting to the reader to comment on which covariates had a larger effect on gene expression. 

We appreciate the reviewer’s comments and have elaborated in greater details the importance of each included covariate on gene expression across the two brain regions (pg. 6; line 169-172). Briefly, in the stepwise regression model, the importance of a given covariate to affect gene expression was based on the frequency by which each covariate is included in the final model. This additional information is also available in Table S3. 

S1 Table. Probably need better legend. What do the numbers mean for Hemisphere, Neuropathology, Hepathology (#should be hepatology?), Toxicology, Smoking. 

We apologize for the confusion and have clarified the data labeling in Table S1. 

Results.

Figure 3. Significance levels for DE not noted in figure, legend, or text. 

We have added the DE significance levels for each MT gene within the figure (pg. 12; line 324-325).

Hub genes of potential biological significance. This section is less interesting. Are they enriched for relevant biological processes or GWAS hits? Are they sig. DE between cases and controls? Other than their high degree of connectivity within modules enriched for AD, what is there biological significance? Or maybe integrate into another results section? 

We appreciate the reviewer’s comments and have included the GO biological processes enrichment for just the hub genes in Table S6. Generally, in network analyses, due to their high connectivity, hub genes play an important role in maintaining the integrity of the biological network. Furthermore, the hub genes are the strongest contributors to summarized ME expression and are also considered the most important genes for future downstream, more functional, studies (pg. 13; line 334-335). Therefore, not surprisingly, we see consistent enrichment of the same GO terms between the two tests, i.e., when only the hub genes are tested versus the entire module (pg.12; 319-322). For these reasons, we also focused our eQTL analysis on the hub genes from AD significant modules only.

Figure 5. Units of expression not clearly described in the legend.

We have changed the legend to clarify the x and y-axis for each plot in Figure 5 (pg. 16; line 406).

Discussion.

Further discussion of the interesting finding of cilium assembly in the NAc but not in the PFC may be warranted. Adult neurogenesis in the NAc? Neurogenesis of medium spiny neurons in the nucleus accumbens continues into adulthood and is enhanced by pathological pain. García-González et al., Mol. Psych. 2020. PMID: 32612250. 

We welcome the suggestion and have included additional discussion on the implications of cilium assembly within the context of neurogenesis in the NAc (pg. 18; line 458-469)

Reviewer #2: The report by Vornholt et al. is a descriptive study that utilizing genome-wide mRNA and miRNA expression from postmortem brain of AD cases compared to matched controls. Gene co-expression networks, mRNA/miRNA interactions, and eQTL analyses were used to identify expression patterns that are uniquely and differently dysregulated in the PFC and NAc. The analytical pipelines are well described and mostly robust.

The Methods indicates 42 cases and 42 controls. Then after matching, 72 total cases. Table S1 shows demos for 70 cases (1 case was a statistical outlier). It is not clear why the initial 12 cases were excluded from the original group (84). 

We appreciate the reviewer’s comments and have clarified this confusion. Essentially, we began with 82 samples (41 cases and 41 controls). These samples were then matched for available covariates presented on pg. 5 lines 140-141, leaving a total of 36 matched cases-controls for downstream analysis. Once the expression data were normalized, one final subject (a case) was removed due to presenting itself as a statistical outlier.

It is not clear from the beginning of the Methods section that the NAc data were reported previously by the same group. Utilization of the previously published data was clarified in the first section of the Results but needs to be outlined clearly in the Methods. Also, for clarity, any annotation updates and data handling differences for the dataset should be listed since the data were published in 2015. 

We have clarified in the methods that the raw data generated in NAc were collected and normalized as part of the previous study. However, considering that both brain regions were assayed on the same microarray platform, and to maintain the consistency of our normalization and analytical pipelines, we opted to re-analyze the expression data from the NAc together with PFC using the same analytical approach (see pg. 6, 166-168). 

There is significant concern about the within subject RNA quality differences for each of the brain regions. A major goal of the reported work was to identify overlapping and unique expression differences in PFC and NAc. There are clear differences in RNA quality between the 2 brain regions. This is likely degradation due to the sample processing at different times. The authors argue reliable data can be obtained from RINs<=4; however, the impact of RNA degradation on downstream expression analysis can be significant, especially when trying to compare multiple brain regions from the same individual. The cross-region analysis is likely compromised.

We agree with the reviewer’s comment. RNA quality and how it may affect subsequent gene expression analyses has been among the most tackled problems in the postmortem brain research area. While generally there has been a positive relationship between RIN and detection of gene expression differences, studies have also shown that reliable gene expression data can also be obtained from RNA with lower RINs [1, 2]. (pg. 20-21; line 529-533). It is also important to note that differences in RINs between cases and controls has a much more detrimental role than low RINs itself [3]. While the RINs in the PFC regions were lower, we did not detect significant differences in RIN scores between cases and controls. Finally, while we acknowledge that the lower PFC RINs may affect our detection power due to our careful analytical design to adjust for the impact of RIN on gene expression, the robustness of our identified differentially expressed genes will remain. 

The major components of the study (descriptive statistics, WGCNA analysis, network preservation, mRNA/miRNA interactions) are potentially interesting; however, there is not a well-integrated overview or thematic model presented from consolidated findings. It is especially hard to understand exactly what the reader will gather from complex mRNA/miRNA interactions. 

We have included additional details (pg. 4; line 117-126 and pg.16; line 418-423) to address this comment in order to provide a more meaningful interpretation of the data with respect to alcohol dependence. 

Functional validation of key findings would significantly strengthen the study. At a minimum, testable hypotheses should be discussed based on the integrated analyses.

We appreciate the reviewer’s comment. The main focus of our study is to compare the gene expression profiles between two critical for alcohol addiction cortical and subcortical brain regions. We have, however, discussed in greater detail future in vivo experiments using animal models and/or neuronal cell cultures, as well as expanding on the methodological framework of this study by increasing postmortem brain sample sizes as well as include additional comparisons from other brain regions. (pg.21, line 538-544)

References:

[1] Weis S, Llenos IC, Dulay JR, Elashoff M, Martínez-Murillo F, Miller CL. Quality control for microarray analysis of human brain samples: The impact of postmortem factors, RNA characteristics, and histopathology. Journal of Neuroscience Methods 2007; 165: 198–209.

[2] Stan AD, Ghose S, Gao X-M, Roberts RC, Lewis-Amezcua K, Hatanpaa KJ, et al. Human postmortem tissue: What quality markers matter? Brain Research 2006; 1123: 1–11.

[3] Gallego Romero I, Pai AA, Tung J, Gilad Y. RNA-seq: impact of RNA degradation on transcript quantification. BMC Biology 2014; 12: 42.

---

## [Decision Letter · Decision Letter 1]

19 Nov 2020

PONE-D-20-30169R1

Network preservation reveals shared and unique biological processes associated with chronic alcohol abuse in NAc and PFC.

PLOS ONE

Dear Dr. Vornholt,

Thank you for resubmitting your manuscript to PLOS ONE. After careful consideration, we feel that it has merit but does not fully meet PLOS ONE’s publication criteria as it currently stands. Therefore, we invite you to submit a revised version of the manuscript that addresses the points raised during the review process.

Specifically, there is still a concern regarding the low RINs. As one of the reviewers suggests, there could be  more quantitative arguments on this issue. In addition, I could not find the database used for sample access.

We look forward to receiving your revised manuscript.

Kind regards,

Andrey E Ryabinin, Ph.D.

Academic Editor

PLOS ONE

Reviewers' comments:

Reviewer's Responses to Questions

**Comments to the Author**

1. If the authors have adequately addressed your comments raised in a previous round of review and you feel that this manuscript is now acceptable for publication, you may indicate that here to bypass the “Comments to the Author” section, enter your conflict of interest statement in the “Confidential to Editor” section, and submit your "Accept" recommendation.

Reviewer #1: All comments have been addressed

Reviewer #2: (No Response)

2. Is the manuscript technically sound, and do the data support the conclusions?

Reviewer #1: Yes

Reviewer #2: Partly

3. Has the statistical analysis been performed appropriately and rigorously? 

Reviewer #1: Yes

Reviewer #2: I Don't Know

4. Have the authors made all data underlying the findings in their manuscript fully available?

Reviewer #1: Yes

Reviewer #2: No

5. Is the manuscript presented in an intelligible fashion and written in standard English?

Reviewer #1: Yes

Reviewer #2: Yes

6. Review Comments to the Author

Reviewer #1: (No Response)

Reviewer #2: There is lingering concern about the impact of the low RIN samples on the analysis. The authors argue that RIN values are consistently low within PFC (thus impacting cases and controls similarly); however, the concern from the initial review concerned a major goal of the work--to identify overlapping and unique expression differences in PFC and NAc. At a minimum, the proportion of variance explained by the known covariates such as age, gender, RIN and PMI should be reported using violin plots or other graphics. Linear mixed models can then be used to quantify the contribution of multiple sources of variation and identify any covariates that should be corrected in the final analysis.

7. PLOS authors have the option to publish the peer review history of their article (what does this mean?). If published, this will include your full peer review and any attached files.

Reviewer #1: No

Reviewer #2: No

---

## [Author Response · Author response to Decision Letter 1]

22 Nov 2020

Response to Reviewers (Revision 2)

Editor Comments:Specifically, there is still a concern regarding the low RINs. As one of the reviewers suggests, there could be more quantitative arguments on this issue. In addition, I could not find the database used for sample access.

We appreciate the editor’s comments and have uploaded our data to GEO on NCBI. The mRNA and miRNA expression data for the NAc was uploaded as part of a previous study (GSE62699). The PFC mRNA and miRNA data is uploaded and currently being processed, but for the sake of expedience we are resubmitting before obtaining an accession number. We will provide the required accession number once it is released to us. 

Reviewer #2: There is lingering concern about the impact of the low RIN samples on the analysis. The authors argue that RIN values are consistently low within PFC (thus impacting cases and controls similarly); however, the concern from the initial review concerned a major goal of the work--to identify overlapping and unique expression differences in PFC and NAc. At a minimum, the proportion of variance explained by the known covariates such as age, gender, RIN and PMI should be reported using violin plots or other graphics. Linear mixed models can then be used to quantify the contribution of multiple sources of variation and identify any covariates that should be corrected in the final analysis.

We appreciate the reviewer’s comment. As suggested by the reviewer, we have provided a graphical representation (Tables S3) of the contribution of each covariate impact on gene expression. We have also provided further justification on using the bidirectional stepwise regression to correct for the covariates’ effect (Pg. 6 Line 172 & 173) on gene expression (Pg. 9 Line 236-241). Furthermore, while the use of linear mixed models is well developed to adjust for hidden covariates effect in gene associations studies, their applicability in gene expression studies might be less evident (for example see Yao at al. (2019)). We also note that by using stepwise regression, the covariates’ effect is assessed individually on each gene, allowing for the inclusion of only the most relevant covariates, thus minimizing model overfitting. Finally, the validity of our model to correctly capture and adjust for the RIN’s effect on gene expression is reflected by the increased number of times RIN is incorporated as a covariate in the regression models assessing gene expression in PFC vs NAc (See Table S3). Therefore, we are confident in our ability to control for the RIN or any other covariates’ effect in our analyses. 

Yao C, Joehanes R, Johnson AD, Huan T, Liu C, Freedman JE, et al. Dynamic Role of trans Regulation of Gene Expression in Relation to Complex Traits. Am J Hum Genet 2017; 100: 571–580.

---

## [Editor Report · Decision Letter 2]

30 Nov 2020

Network preservation reveals shared and unique biological processes associated with chronic alcohol abuse in NAc and PFC.

PONE-D-20-30169R2

Dear Dr. Vornholt,

We’re pleased to inform you that your manuscript has been judged scientifically suitable for publication and will be formally accepted for publication once it meets all outstanding technical requirements.

Kind regards,

Andrey E Ryabinin, Ph.D.

Academic Editor

PLOS ONE
---

## [Editor Report · Acceptance letter]

7 Dec 2020

PONE-D-20-30169R2 

Network preservation reveals shared and unique biological processes associated with chronic alcohol abuse in NAc and PFC. 

Dear Dr. Vornholt:

I'm pleased to inform you that your manuscript has been deemed suitable for publication in PLOS ONE. Congratulations! Your manuscript is now with our production department. 

Kind regards, 

on behalf of

Dr. Andrey E Ryabinin 

Academic Editor

PLOS ONE